# Carbohydrate Intake Does Not Counter the Post-Exercise Decrease in Natural Killer Cell Cytotoxicity

**DOI:** 10.3390/nu10111658

**Published:** 2018-11-04

**Authors:** Laurel M. Wentz, David C. Nieman, Jennifer E. McBride, Nicholas D. Gillitt, Leonard L. Williams, Renaud F. Warin

**Affiliations:** 1Department of Nutrition and Health Care Management, Appalachian State University, Boone, NC 28608, USA; wentzlm@appstate.edu; 2Human Performance Laboratory, Appalachian State University, North Carolina Research Campus, Kannapolis, NC 28081, USA; mcbrideje@appstate.edu; 3Dole Nutrition Research Laboratory, North Carolina Research Campus, Kannapolis, NC 28081, USA; nicholas.gillitt@dole.com (N.D.G.); renaudwarin@gmail.com (R.F.W.); 4Center for Excellence in Post-Harvest Technologies, North Carolina Research Campus, Kannapolis, NC 28081, USA; llw@ag.ncat.edu

**Keywords:** Immunity, leukocyte, lymphocyte, flow cytometry, glucose, exercise

## Abstract

In a study using a randomized crossover approach, cyclists (*n* = 20, overnight fasted) engaged in three 75 km time trials while ingesting water (WAT) or carbohydrate (0.2 g/kg every 15 min) from bananas (BAN) or a 6% sugar beverage (SUG). Blood samples were collected pre-exercise and 0 h, 1.5 h, and 21 h post-exercise and analyzed for natural killer (NK) cytotoxicity activity (NKCA) using pure NK cell populations. The two carbohydrate trials (BAN, SUG) compared to WAT were associated with higher post-exercise glucose and lower cortisol, total blood leukocyte, neutrophil, and NK cell counts (interaction effects, *p* < 0.001). The immediate post-exercise increase in NK cell counts was higher in WAT (78%) compared to BAN (32%) and SUG (15%) trials (*p* ≤ 0.017). The 1.5 h post-exercise decrease in NK cell counts did not differ after WAT (−46%), BAN (−46%), and SUG (−51%) trials. The pattern of change in post-exercise NKCA differed between trials (*p* < 0.001). The 1.5 h post-exercise decreases in NKCA were 23%, 29%, and 33% in the WAT, BAN, and SUG trials, respectively, but trial contrasts did not differ significantly. Carbohydrate ingestion from BAN or SUG attenuated immediate post-exercise increases in leukocyte, neutrophil, and NK cell counts, but did not counter the 1.5 h decreases in NK cell counts and NKCA.

## 1. Introduction

Natural killer (NK) cells are an essential element of the innate immune system and have the capacity to quickly recognize and eliminate abnormal cells through cell-to-cell contact and without prior activation. Prior studies indicate that intense and prolonged exercise transiently decreases NK cell function, creating a post-exercise window of immune suppression that may increase the risk of infection [1]. Intense exercise lasting longer than 90 min leads to a 35–60% reduction in NK cell cytotoxic activity (NKCA) for up to 6 h after exercise [1,2]. Carbohydrates consumed during exercise attenuate the post-exercise immune response by increasing plasma glucose and insulin while reducing stress hormones (e.g., epinephrine, cortisol) [3]. Compared to water, carbohydrates have been shown to suppress the post-exercise release of total leukocytes, neutrophils, monocytes, and lymphocytes, including NK cells [1,4]. Notably, bananas have been shown to match a commercial carbohydrate solution’s ability to attenuate the post-exercise inflammatory response, with the added benefit of increasing dopamine while decreasing COX-2 mRNA expression [5,6]. Many other banana metabolites increase in the circulation following acute ingestion, and these may have an influence on post-exercise immune response [6]. While carbohydrates have been shown to reduce NK cell counts following exercise, they have not been found to counter the exercise-induced decrease in NKCA, and polyphenol-rich fruits (e.g., bananas) have not been tested to date.

This interpretation of existing data has been questioned, however, due to methodological issues. We have recently developed a highly reliable assay for NKCA in fresh human blood samples to mitigate analysis issues related to the use of Ficoll gradients, viability and stability, and high throughput [7]. This NK cell method utilizes a magnetic-based cell sorter to generate a pure NK cell population and an imaging flow cytometer with an optimized target-to-effector (T:E) ratio that improves the detection of exercise-induced effects on NKCA. Previous research to detect NKCA has relied on mathematical equations to calculate cytotoxicity on a per-NK-cell basis.

The purpose of this study was to compare NKCA responses to 75 km of cycling in active individuals ingesting carbohydrate from Cavendish bananas or a 6% sugar beverage with water using the optimized targeted NKCA assay. This study advances the current state of the literature by using a pure NK cell population and shows that a decline in NKCA occurs following intensive and prolonged exercise, but not to the extent previously reported. Carbohydrate supplementation by either banana fruit or the 6% sugar beverage did not counter post-exercise reductions in NKCA, and this finding is consistent with prior publications despite the use of pure NK cell populations.

## 2. Materials and Methods

### 2.1. Participants 

The metabolomics data from this study have been published previously [6], and the data in this paper summarize the NK responses to both exercise stress and carbohydrate intake. Participants included 20 male and female cyclists (ages 22–50 years) who regularly competed in road races (category 1 to 5) and were capable of cycling for 75 km at a race pace in a laboratory setting. Participants maintained their typical training regimen and avoided the use of vitamin and mineral supplements, herbs, and medications during data collection. Participants voluntarily signed the informed consent, with study procedures approved by the university’s Institutional Review Board (Trial registration: ClinicalTrials.gov, U.S. National Institutes of Health, identifier: NCT02994628).

### 2.2. Research Design

This study utilized a randomized crossover approach, and participants engaged in three 75 km cycling time trials while ingesting water only (WAT), Cavendish bananas (BAN), or a 6% carbohydrate beverage (SUG), separated by 2 weeks each (no blinding). Prior to the cycling time trials, maximal power, oxygen consumption, ventilation, and heart rate were measured during a graded exercise test (25 watt increase every 2 minutes, starting at 150 watts) with the Cosmed Quark CPET metabolic cart (Cosmed, Rome, Italy) and the Lode cycle ergometer (Lode Excalibur Sport, Lode B.V., Groningen, The Netherlands). Body composition was measured with the Bod Pod body composition analyzer (Life Measurement, Concord, CA, USA). Demographic and training histories were acquired with questionnaires.

Participants were asked to reduce the volume of their exercise training as if preparing for a race prior to each 75 km cycling time trial. Participants agreed to ingest a moderate-carbohydrate diet during the 3 day period prior to each exercise session using a food list restricting high-fat foods, and to record intake in food logs. Nutrient intake was assessed using the Food Processor v. 11.1 software system (ESHA Research, Salem, OR, USA).

For each trial, participants reported to the Human Performance Laboratory at 06:45 in an overnight fasted state (no food or beverages other than water for at least 9 h) and provided a pre-exercise blood sample. Participants then ingested 5 mL/kg water only, or water with 0.4 g/kg carbohydrate from Cavendish bananas (ripeness stage 5 or 6), or the 6% sugar beverage in accordance with the randomized schedule. At approximately 07:15, the cyclists warmed up and cycled for 75 km at race-pace intensity using their own bicycles on CompuTrainer Pro Model 8001 trainers (RacerMate, Seattle, WA, USA). The CompuTrainer MultiRider software system (version 3.0, RacerMate, Seattle, WA, USA) was used to simulate a moderately difficult mountainous 75 km course. Power output in watts was continuously monitored, with heart rate recorded every 30 min. Oxygen consumption and ventilation were measured during 2 level portions of the race course (16 and 56 km) using the Cosmed Quark CPET metabolic cart. Every 15 min, participants consumed 3 mL/kg water, or water with 0.2 g/kg carbohydrate from bananas, or the 6% sugar beverage. No other beverages or food were allowed during the cycling time trials and 1.5 h recovery. Blood samples were taken via venipuncture immediately, 1.5 h, and 21 h post-exercise after completing each of the 75 km time trials. The 21 h post-exercise samples were obtained from participants at ~07:00 in an overnight fasted state. All blood samples were centrifuged, aliquoted, and stored at −80 °C until analysis. The 3 trials were separated by 2 weeks, after which participants crossed over to the next randomized condition and repeated all procedures.

### 2.3. Analytical Methods

#### 2.3.1. Complete Blood Count, Glucose, Cortisol

Complete blood counts (CBCs) with a white blood cell (WBC) differential were performed using a Coulter Ac⋅T^TM^ 5diff Hematology Analyzer (Beckman Coulter Inc., Miami, FL, USA). Exercise-induced shifts in plasma volume were calculated using the equation of Dill and Costill [7]. Plasma glucose was measured using the YSI 2300 STAT Plus Glucose and Lactate analyzer (YSI Life Sciences, Yellow Springs, OH, USA). Plasma cortisol was measured using an ultraperformance liquid chromatography–tandem mass spectrometry (UPLC-MS/MS) platform, a Waters Acquity UPLC, and a Thermo Scientific Q-Exactive mass spectrometer (Thermo Scientific, Waltham, MA, USA).

#### 2.3.2. Natural Killer Cell Assay

Flow cytometric analysis of natural killer cell cytotoxicity activity (NKCA) was measured in whole blood samples using the procedures of McBride et al. [8]. In brief, immediately following each blood draw, a pure population of NK cells was isolated from 1 mL whole blood labeled with CD56+ MicroBeads and processed through the autoMACS Pro Separator (Miltenyi Biotech, Bergisch Gladbach, Germany). NK cell counts were obtained using a hemocytometer. Target K562 cells (American Type Culture Collection, Rockville, MD, USA) were stained with 3,3′-diotadecyloxacarbocyanine perchlorate (DiO) and/or propidium iodide (PI) dye, as follows: resuspended DiO- and PI-labeled K562 cells (double positive), DiO-labeled K562 cells (DiO only), and PI-labeled K562 cells (PI only). The optimized ratio of NK effector cells (E) to K562-DiO labeled target cells (T) was tested and set at 1:5 E:T. Target and effector cells were then combined in 500 μL of NK cell medium without interleukin-2 (IL-2) and 2-mercaptoethanol (2-ME) (incomplete NK cell culture medium), and incubated for 2 h at 5% CO_2_/37 °C. Spontaneous samples were prepared with DiO-labeled K562 cells only in the incomplete NK cell culture medium.

After incubation, NK cytotoxicity was measured using the Amnis ImageStream^®^X Mark II Imaging Flow Cytometer (EMD Millipore, Burlington, MA, USA) (Figure 1). Controls were analyzed prior to experimental samples in the following order: double positive, DiO only, and PI only. Experimental data were processed using IDEAS software (application version 6.2.64.0). The percentage of dead targets in the spontaneous sample and experimental samples was determined using the following formula: % dead targets in sample = (#dead targets × 100)/(#live targets + #dead targets).

NK cytotoxicity was determined using the following formula:% cytotoxicity = [(Experimental dead − Spontaneous dead)/(100 − Spontaneous dead)] × 100.(1)

### 2.4. Statistical Procedures

Data are presented as mean ± standard error (SE). Immune data were analyzed using 3 (trial) × 4 (time) repeated-measures ANOVA, within-participants design, with IBM SPSS Statistics for Windows, Version 24.0 (IBM Corp., Armonk, NY, USA). Changes over time within trials were contrasted between trials using Bonferroni-corrected paired t-tests. Statistical differences were accepted when the *p*-value was ≤0.017. The study participant number (*n* = 20) provided 84% power to detect a difference with an effect size of 0.7 at alpha 0.05 using 2-sided paired *t*-tests.

## 3. Results

The analysis included 20 cyclists (14 males, 6 females) who successfully adhered to all aspects of the study design. The male and female cyclists did not differ in age (37.1 ± 2.5 and 43.7 ± 2.2 years, respectively, *p* = 0.126), training volume (118 ± 13.6 and 136 ± 24.1 km/wk, *p* = 0.520), body composition (19.5 ± 1.3% and 18.8 ± 1.9% fat, *p* = 0.763), or VO_2max_ (47.0 ± 1.5 and 46.5 ± 2.8 mL·kg^.−1^ min^−1^, *p* = 0.861). Data for the male and female cyclists were combined for all analyses in this paper. Three-day food records collected before each of the three 75 km cycling time trials revealed no significant trial differences in energy, carbohydrate, and micronutrient intake (data not shown).

Performance times (180 ± 4.8, 176 ± 4.5, 178 ± 3.7 min), absolute oxygen consumption (2.53 ± 0.89, 2.62 ± 1.06, 2.45 ± 0.93 L/min), heart rates (142 ± 2.6, 140 ± 3.4, 143 ± 2.8 beats/min), and plasma volume decreases (−12.2 ± 1.2%, −8.1 ± 1.0%, −11.1 ± 1.5%) did not differ significantly (all *p* > 0.05) during the BAN and SUG trials compared to the WAT condition. Plasma glucose and cortisol data have been published previously [6]. In brief, plasma glucose was significantly elevated during the first 1.5 h post-exercise in the BAN and SUG trials compared to the water trial, with a significant rebound in plasma glucose in the WAT condition following lunch (consumed after the 1.5 h post-exercise blood collection) (interaction effect, *p* < 0.001). Change in plasma cortisol levels was significantly lower during the first 1.5 h of recovery from exercise with BAN (42%) and SUG ingestion (32%) compared to WAT (interaction effect, *p* < 0.001).

The pattern of change in post-exercise total blood leukocyte, neutrophil, lymphocyte, and monocyte counts was significantly different between trials (all interaction effects, *p* < 0.01), with lower total leukocyte and neutrophil counts during the first 1.5 h recovery for the BAN and SUG trials compared to WAT (Table 1). The pattern of change in post-exercise NK cell counts was significantly different between trials (interaction effect, *p* < 0.001), with the immediate post-exercise increase after the WAT trial (78%) significantly higher than after the BAN (32%) and SUG (15%) trials (both contrasts, *p* ≤ 0.017) (Figure 2). The 1.5 h post-exercise decrease in NK cell counts did not differ significantly after the WAT (−46%), BAN (−46%), and SUG (−51%) trials. The pattern of change in post-exercise NKCA differed between trials (interaction effect, *p* < 0.001) (Figure 3). The 1.5 h post-exercise decreases in NKCA were 23%, 29%, and 33% in the WAT, BAN, and SUG trials, respectively, but trial contrasts did not differ significantly.

## 4. Discussion

This randomized crossover study utilized a pure NK cell population to analyze changes in NKCA following 75 km cycling trials with carbohydrate supplementation from a 6% sugar beverage or banana fruit compared to water. Our data show that carbohydrate supplementation attenuated the immediate post-exercise elevations in NK cell counts but this did not translate into improvements in cytotoxicity across recovery. NKCA decreased post-exercise in all treatments, and neither the banana phenolics nor the sugar from carbohydrates attenuated the post-exercise NKCA response. There was a small yet statistically significant difference in pattern of change in NKCA across treatments, although it was not significant at any specific time point, and differences were too small to be of clinical significance. Using advanced methodologies, this study shows that carbohydrates did not counter NKCA but did attenuate immediate post-exercise increases in total leukocyte, neutrophil, and NK cell counts, supporting an effect of carbohydrates on immune cell counts but not cytotoxicity during endurance exercise.

Previous studies have reported significant shifts in NK cell counts and cytotoxicity following endurance exercise, and the data in this study confirm these findings, albeit at a lower magnitude. NKCA assay methodological differences, especially the use of pure NK cell populations in the current study, may explain the contrast between studies. Early research showed that NKCA was reduced by 60% at 1 h [2] and 58% at 1.5 h [1] post-exercise, whereas our data support less pronounced declines of 23%, 29%, and 33% in the WAT, BAN, and SUG trials, respectively, at 1.5 h post-exercise, with no significant differences between trials. These data are consistent with previous findings showing that carbohydrate ingestion does not effectively counter the post-exercise decrease in NKCA [1,2,9]. Henson et al. [10], for example, found that NKCA was not lower with carbohydrate supplementation compared to water placebo after 3 h recovery from 2.5 h of endurance exercise at 75% VO_2max_. Our data show that NKCA at 1.5 h post-exercise was slightly lower with ingestion of SUG, but this was not significantly different from the BAN or WAT trials. Some but not all researchers have found that NKCA responses to exercise parallel changes in NK cell counts [1,2,9,10]. In this study, the pattern of change in NK cell counts differed from NKCA, showing an immediate post-exercise spike that was greater with WAT (78%) vs. carbohydrate (32% in BAN and 15% in SUG trials), although all treatments dropped 46–51% below baseline by 1.5 h post-exercise with no significant differences between trials. Most research has supported the effect of carbohydrate on attenuating the post-exercise spike in NK cells. Following 2.5 h of endurance exercise, carbohydrate supplementation reduced the post-exercise NK cell response to 23–32% above baseline vs. 81–91% increase with water across running and cycling trials [10,11]. Research has consistently shown that by 1.5 h post-exercise, NK cell counts decline to 52–65% below baseline independent of carbohydrate supplementation, which supports the findings in this study [1,10,11]. However, not all research supports a significant effect of carbohydrates on NK cell concentrations [2,9]. Nieman et al. [2] showed that carbohydrates affected neither NK cell counts nor NKCA following 2 h of cycling compared to water. Likewise, McFarlin et al. [9] found that carbohydrates did not alter NK cell patterns or NKCA following 1 h of cycling; however, carbohydrates did enhance the *in vitro* NK cell responsiveness to IL-2.

Carbohydrate ingestion alters multiple immune responses to exercise, as evidenced by attenuation of the immediate post-exercise increases in total leukocyte, neutrophil, and NK cell counts (Table 1). Consuming carbohydrates from 6–8% sugar beverages or sugar-dense fruits such as banana (with water) during prolonged vigorous exercise reduces circulating concentrations of epinephrine and cortisol, resulting in reduced inflammation and modulation of immune cell responses [3]. Bananas are rich in phenolics that increase amino acid and xenobiotic metabolites and have been shown to increase antioxidant markers [5,6]. In this study, bananas matched but did not exceed a sugar beverage in modulating immune cells, and neither had a significant effect on NKCA. NK cell concentration is more responsive to exercise than other lymphocytes, as evidenced by an exercise-induced rapid release of cells into the bloodstream followed by a rapid efflux once activity has ceased [12]. Exercise causes a surge in circulating epinephrine, which in turn mobilizes NK cells into the bloodstream via activation of β-adrenergic receptors [13]. The cytokine IL-6 is also released during exercise and may play a role in redistribution of NK cells from the bloodstream to tissues [14]. NK cells egress quickly from the bloodstream [15], and by 1.5 h post-exercise have dropped below pre-exercise concentrations (Figure 2). The exercise-induced surge in epinephrine is short-lived as well, as previous work has shown that, despite carbohydrate attenuation immediately post-exercise, epinephrine concentrations are similar with and without carbohydrate supplementation and return to near baseline by 1–1.5 h post-exercise, showing a transient post-exercise effect [2,10,11]. The rapid release of epinephrine and NK cells immediately following exercise, followed by the rapid redistribution from the bloodstream, may explain why carbohydrates did not counter reductions in NKCA. By 1.5 h post-exercise, when NCKA reached its nadir, epinephrine and NK cell concentrations had returned to or decreased from baseline. Thus, the increases in epinephrine and NK cell concentrations do not translate to increases in cytotoxicity, as evidenced by no effect on NKCA.

## 5. Conclusions

Our data show that carbohydrate supplementation from a 6% sugar beverage or banana fruit during 75 km cycling time trials attenuated the immediate post-exercise increases in total leukocyte, neutrophil, and NK cell counts, but did not counter the 1.5 h decreases in NK cell counts and NK cell cytotoxicity activity against target cells. These findings support previous literature showing that carbohydrates do not modulate exercise-induced reductions in NKCA using a pure NK cell population and targeted assay.

## Figures and Tables

**Figure 1 nutrients-10-01658-f001:**
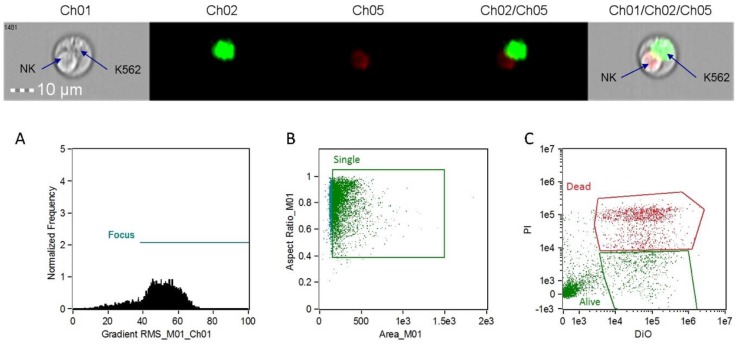
Image data collected from the ImageStream^®^X Mark II Imaging Flow Cytometer. In the middle section, target cells (K562) are labeled in green (diotadecyloxacarbocyanine perchlorate, DiO) and dying cells are labeled in red (propidium iodide, PI). The images on the ends represent a doublet event showing an apoptotic natural killer (NK) cell and a live K562 target. (**A**) Focus cell analysis histogram; (**B**) single cell analysis scatterplot; (**C**) target cell staining analysis scatterplot.

**Figure 2 nutrients-10-01658-f002:**
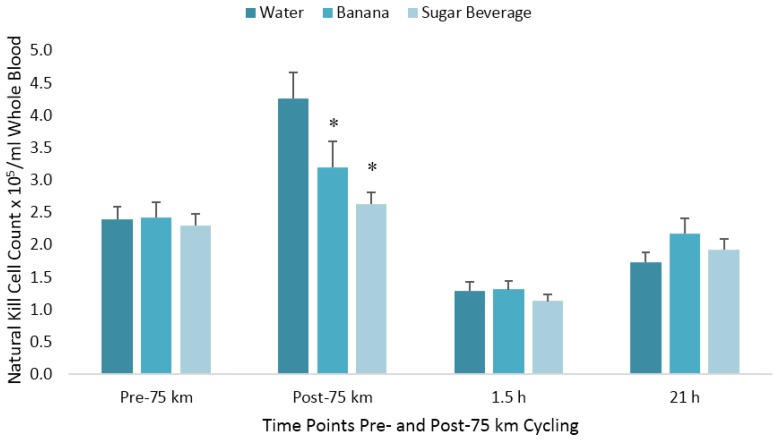
Changes in natural killer cell counts per milliliter of whole blood following 75 km cycling in *n* = 20 cyclists (immediately, 1.5 h, and 21 h post-exercise). * *p* ≤ 0.017 compared to the change from pre-exercise in the water condition. Vertical lines represent the standard errors. Time effect, *p* < 0.001; time × trial effect, *p* < 0.001.

**Figure 3 nutrients-10-01658-f003:**
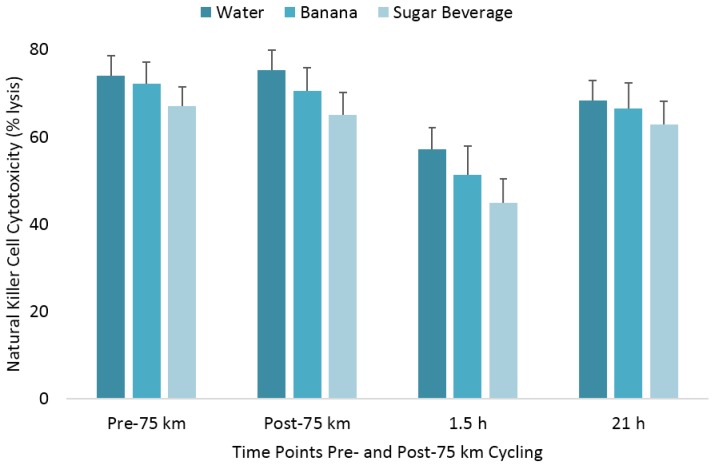
Change in natural killer cell cytotoxicity following 75 km cycling in *n* = 20 cyclists (immediately, 1.5 h, and 21 h post-exercise). The changes from pre-exercise in the banana and sugar beverage trials did not differ from the water trial. Vertical lines represent the standard errors. Time effect, *p* = 0.014; time × trial effect, *p* < 0.001.

**Table 1 nutrients-10-01658-t001:** Change in leukocyte subset counts following 75 km cycling in *n* = 20 cyclists (immediately, 1.5 h, and 21 h post-exercise).

Variable	Water	Cavendish Banana	Sugar Beverage	*p*-Values (Time Effect; Interaction Effect)
**Total blood leukocytes (10^9^/L)**				
Pre-exercise	5.15 ± 0.3	5.16 ± 0.4	5.24 ± 0.4	<0.001; <0.001
Immediate post-exercise	15.6 ± 1.3	10.5 ± 0.7 *	9.70 ± 0.6 *	
1.5 h post-exercise	12.2 ± 1.0	9.11 ± 0.5 *	9.07 ± 0.6 *	
21 h post-exercise	5.72 ± 0.4	5.09 ± 0.3	5.13 ± 0.3	
**Neutrophil count (10^9^/L)**				
Pre-exercise	2.50 ± 0.2	2.48 ± 0.2	2.71 ± 0.3	<0.001; <0.001
Immediate post-exercise	11.7 ± 1.1	7.34 ± 0.6 *	6.71 ± 0.5 *	
1.5 h post-exercise	10.0 ± 0.9	6.82 ± 0.5 *	6.86 ± 0.5 *	
21 h post-exercise	3.05 ± 0.3	2.49 ± 0.2 *	2.66 ± 0.2 *	
**Lymphocyte count (10^9^/L)**				
Pre-exercise	1.98 ± 0.1	2.01 ± 0.1	1.87 ± 0.1	<0.001; 0.009
Immediate post-exercise	2.53 ± 0.1	2.28 ± 0.2	2.11 ± 0.1	
1.5 h post-exercise	1.26 ± 0.1	1.55 ± 0.1	1.46 ± 0.1	
21 h post-exercise	1.98 ± 0.1	1.94 ± 0.1	1.78 ± 0.1	
**Monocyte count (10^9^/L)**				
Pre-exercise	0.43 ± 0.1	0.43 ± 0.1	0.42 ± 0.1	<0.001; 0.006
Immediate post-exercise	1.01 ± 0.1	1.02 ± 0.1	1.12 ± 0.1 *	
1.5 h post-exercise	0.71 ± 0.1	0.80 ± 0.1 *	0.80 ± 0.1 *	
21 h post-exercise	0.46 ± 0.1	0.50 ± 0.1	0.50 ± 0.1 *	

* *p* < 0.017 compared to the change from pre-exercise in the water condition.

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
