# Peer review of "Carbohydrate Intake Does Not Counter the Post-Exercise Decrease in Natural Killer Cell Cytotoxicity"

_nutrients, 2018, doi:10.3390/nu10111658_

Reviewer 1 Report

The results are interesting. However, the participants and research design in this study were the same as the Nieman’s article published in 2018, (Nieman, D.C.; Gillitt, N.D.; Sha, W.; Esposito, D.; Ramamoorthy, S. Metabolic recovery from heavy exertion following banana compared to sugar beverage or water only ingestion: A randomized, crossover trial. PLoS One 2018, 13, e0194843, doi:10.1371/journal.pone.0194843) and the metabolomics data showing in this study have been published previously. Thus, in my opinion, authors should remove the data in this manuscript which have been published. The concerns showing as following.

 Major concerns

Remove the results that have been published and reorganized this manuscript.

The materials and methods section (2.3.1) present the analytical methods of glucose and cortisol, however, there is no glucose and cortisol data presented.

 Minor concern

Error typing in Figure 2, Y-axis. Natural killer cell count x105/ml -à 105

 Author Response

Response to Reviewer 1 Comments

 Point 1: The results are interesting. However, the participants and research design in this study were the same as the Nieman’s article published in 2018, (Nieman, D.C.; Gillitt, N.D.; Sha, W.; Esposito, D.; Ramamoorthy, S. Metabolic recovery from heavy exertion following banana compared to sugar beverage or water only ingestion: A randomized, crossover trial. PLoS One 2018, 13, e0194843, doi:10.1371/journal.pone.0194843) and the metabolomics data showing in this study have been published previously. Thus, in my opinion, authors should remove the data in this manuscript which have been published. The concerns showing as following.

Response 1: Thank you for reviewing the manuscript and for your comments.

Major concerns

Point 2: Remove the results that have been published and reorganized this manuscript.

 Response 2: Data on leukocyte subsets has not been published previously nor has data on NK cells. Descriptive data are briefly presented in the results text to show characteristics of the study participants. Reviewer 3 asked for descriptive data on the participants to be included. A summary of glucose and cortisol changes with treatment conditions and exercise is included to support mechanism for changes in immune cells. It has been noted when data have previously been published.

 PARAGRAPH IN LINES 160-169 on glucose

Point 3: The materials and methods section (2.3.1) present the analytical methods of glucose and cortisol, however, there is no glucose and cortisol data presented.

Response 3: A brief summary of analytical methods are provided in the methods. Glucose and cortisol data were presented in Nieman et al. PLoS ONE 13(3): e0194843. https://doi.org/10.1371/journal.pone.0194843. This citation has been included in the results to refer readers to the data.  A summary of changes in glucose and cortisol across the trials are presented in the second paragraph of the results, lines 163-169.

Minor concern

Point 4: Error typing in Figure 2, Y-axis. Natural killer cell count x105/ml -à 105

Response 4: We have corrected the Figure 2 y-axis label.

Reviewer 2 Report

Wentz et al., have continued the previous work of Nieman and co- workers. While the work is well presented, it appears to add little new information to the field. It is also unclear while some data e.g. changes in plasma cortisol are not shown.

Table 2 contains a typographical error on the y axis.

Author Response

Response to Reviewer 2 Comments

 Point 1: Wentz et al., have continued the previous work of Nieman and co- workers. While the work is well presented, it appears to add little new information to the field. It is also unclear while some data e.g. changes in plasma cortisol are not shown.

 Response 1: Thank you for reviewing the manuscript and for your comments. This manuscript uses a novel method of assessing NK cells, identifying a pure NK population that has not been achieved in previous assays. We did identify some differences from previous literature in the magnitude of changes on NK cells in response to exercise. However, like previous literature, we did not show an effect on carbohydrate on NK cell response.

 A summary of changes in glucose and cortisol across the trials are presented in the second paragraph of the results, lines 163-169. Glucose and cortisol data were presented in Nieman et al. PLoS ONE 13(3): e0194843. https://doi.org/10.1371/journal.pone.0194843.  This citation has been included in the results to refer readers to the original data. 

 Point 2: Table 2 contains a typographical error on the y axis.

Response 2: We have corrected the Figure 2 y-axis label.

Reviewer 3 Report

Thank you for your manuscript - you have done a lot of work to get the trials with sufficient sample size done!

I would like to see more descreptive data on the participants

Could you justify why to use SE and not SD?

Why didnt you report cortisol data?

Are your subjects really athletes?

Author Response

Response to Reviewer 3 Comments

 Point 1: Thank you for your manuscript - you have done a lot of work to get the trials with sufficient sample size done!

Response 1: Thank you for reviewing the manuscript and for your comments. We appreciate the feedback.

Point 2: I would like to see more descreptive data on the participants

Response 2: Detailed descriptive data on the participants has been published in Nieman et al. PLoS ONE 13(3): e0194843. https://doi.org/10.1371/journal.pone.0194843. We have included data on participants age, training volumes, body composition, and Vo2max in the results section (lines 153-159).

Point 3: Could you justify why to use SE and not SD?

Response 3: Data were analyzed using repeated measures. Research supports use of standard error bars for repeated measures, within-subject designs.  

Point 4: Why didnt you report cortisol data?

Response 4: Cortisol data have previously been reported in Nieman et al. PLoS ONE 13(3): e0194843. https://doi.org/10.1371/journal.pone.0194843, and we did not want to repeat publication of result. This citation has been included in the results to refer readers to the data. A summary of changes in cortisol across the trials are presented in the second paragraph of the results, lines 167-169.  

Point 5: Are your subjects really athletes?

Response 5: Subjects were active individuals who regularly competed in road races (category 1 to 5) and were capable of cycling 75-km at race pace in a laboratory setting but were not technically athletes. The term athletes has been removed from the manuscript and replaced with active individuals (line 58-59).

Round  2

Reviewer 1 Report

I have no other comments.